# Comparison between Deep Learning and Conventional Machine Learning in Classifying Iliofemoral Deep Venous Thrombosis upon CT Venography

**DOI:** 10.3390/diagnostics12020274

**Published:** 2022-01-21

**Authors:** Jung Han Hwang, Jae Won Seo, Jeong Ho Kim, Suyoung Park, Young Jae Kim, Kwang Gi Kim

**Affiliations:** 1Department of Radiology, Gil Medical Center, Gachon University College of Medicine, Incheon 21565, Korea; lj3800@naver.com (J.H.H.); hwp20@naver.com (S.P.); 2Department of Health Sciences and Technology, GAIHST, Gachon University, Incheon 21999, Korea; sjw0614@gachon.ac.kr; 3Department of Biomedical Engineering, Gil Medical Center, Gachon University, Incheon 21565, Korea; youngjae@gachon.ac.kr

**Keywords:** deep vein thrombosis, computed tomography, radiomics, deep learning, machine learning

## Abstract

In this study, we aimed to investigate quantitative differences in performance in terms of comparing the automated classification of deep vein thrombosis (DVT) using two categories of artificial intelligence algorithms: deep learning based on convolutional neural networks (CNNs) and conventional machine learning. We retrospectively enrolled 659 participants (DVT patients, 282; normal controls, 377) who were evaluated using contrast-enhanced lower extremity computed tomography (CT) venography. Conventional machine learning consists of logistic regression (LR), support vector machines (SVM), random forests (RF), and extreme gradient boosts (XGB). Deep learning based on CNN included the VGG16, VGG19, Resnet50, and Resnet152 models. According to the mean generated AUC values, we found that the CNN-based VGG16 model showed a 0.007 higher performance (0.982 ± 0.014) as compared with the XGB model (0.975 ± 0.010), which showed the highest performance among the conventional machine learning models. In the conventional machine learning-based classifications, we found that the radiomic features presenting a statistically significant effect were median values and skewness. We found that the VGG16 model within the deep learning algorithm distinguished deep vein thrombosis on CT images most accurately, with slightly higher AUC values as compared with the other AI algorithms used in this study. Our results guide research directions and medical practice.

## 1. Introduction

Venous thromboembolism is the third most common cause of cardiovascular disease worldwide, following coronary artery disease and stroke [1]. Its incidence has dramatically increased over the last two decades [1,2]. Venous thromboembolism occurs in two forms: pulmonary embolism and deep vein thrombosis (DVT). DVT is often associated with recurrent pulmonary embolism and venous thromboembolism [3]. Recurrence rates during the disease course are approximately 20–36% [4,5]. Computed tomography (CT) venography of the lower extremities presents the most common diagnostic tool for ascertaining DVT [4]. CT imaging has evolved from being a primary diagnostic tool to a critical component of individualized precision medicine [6].

Visual analysis of CT scans to diagnose DVT is time-consuming and subjective, whereas computer classification is significantly faster and more objective. Radiomics is a unique tool for clinical diagnosis that extracts mineable high-throughput features from medical images using automated algorithms [7]. Radiomics approaches have been widely applied in CT image interpretation throughout the last decade, and they have demonstrated promising results in differential diagnoses [8]. Although radiomics models operate within an acceptable error range for specific tasks, radiomics-based techniques have a number of drawbacks. Handcrafted radiomic features, for example, are limited to current medical imaging expertise and prior operator experience and knowledge. As a result, these characteristics might not be properly representative [9].

A convolutional neural network (CNN) can extract and learn deep features directly in a data-driven way, as opposed to traditional radiomics methods that rely on handcrafted features [10]. In various deep-running fields, CNN is of high performance in image recognition, and many studies are based on CNNs in medical imaging, including various vision sectors [11,12]. CNN methods perform well in image classification and in the recognition of medical imaging [13]. These architectures have strong potential for enhancing workflow processes in radiology. During the training phase, however, classification labels without lesion information may not provide effective supervised information on the suspected lesion area. This could make it difficult for CNN classification to extract useful information about lesions, especially small lesions, resulting in poor performance.

In this study, we used a large dataset comprising lower-extremity CT venography images to demonstrate the effectiveness of the proposed methodology for the automated classification of DVT lesions. We extensively evaluated the classification performance of two model types (CNN and conventional machine learning). The aim of the present study was to determine the feasibility of classifying DVT upon CT venography using suitable CNN method.

## 2. Materials and Methods

### 2.1. Study Design

This single-center retrospective study was approved by the institutional review board of our local ethics committee. This study was conducted in accordance with the principles of the Declaration of Helsinki. The requirement for written informed consent was waived given the retrospective nature of our study.

In this study, among the artificial intelligence (AI) algorithms, we define the machine-learning, which yields decision with specific function by parsing generated features and requires adjustment of engineer (such as feature selection) for prediction as “conventional machine learning”. The algorithms, which create features and make decisions on its own deep network are defined as “deep learning”. To distinguish the presence or absence of thrombosis in deep veins through image-based analyses, we applied four classifiers using embedded feature selection methods based on conventional machine learning as well as four deep learning algorithms based on CNN. We compared quantitative differences in performance and investigated the best model for classifying blood vessels into two categories. Moreover, we evaluated the likely reason for the obtained results according to each method’s classifier and structure. Figure 1 shows the flowchart of the study design and process.

### 2.2. Data Acquisition and Preprocessing

CT images from 282 patients with deep vein thrombosis and 377 patients with non-deep vein thrombosis (designated as healthy controls) were retrospectively collected from the Gil Medical Center (Incheon, Republic of Korea) in DICOM (digital imaging and communications in medicine) format. Each CT image window setting was set to a window width of 140HU and window level of 400HU for enhancing the contrast between the blood vessel walls and the thrombus. Because the range of pixel spacing was wide (0.695–0.977 mm^2^) and the regions of the deep veins were diminutive, we uniformed the spacing of pixel to a mean value (0.891 mm^2^) of the pixel spacing range. As shown in Figure 2, from this preprocessed data, we selected 812 images of the deep vein region with thrombus from patients with deep vein thrombosis and 812 images of the deep vein region from non-deep vein thrombosis patients. Among the total of 1624 image data (deep vein with thrombus, *n* = 650; healthy deep vein without thrombus, *n* = 650), 976 data and 324 data were used for the training and validation set, respectively. The remaining 324 data from the total, excluding the training and validation set, were used as the testing set to evaluate performance.

### 2.3. Classification Using Conventional Machine Learning

Statistical texture analysis is a method for analyzing the distributions and correlations of brightness values in image pixels using mathematical methods. Statistical-based methods include first-order statistics calculated using feature values based on histograms within the region of interest (ROI) as well as second-order statistics measured from matrices formed based on the frequencies and relationships for each pixel value in the gray level [14]. The first-order statistics and second-order statistics are used for classifying images [13,14]. Therefore, we applied radiomic features of first order statistics and second-order statistics to a conventional machine learning classifier for classifying the images. [15,16]. From a total of 1624 images, we extracted 74 features as follows: 18 from first-order statistics, 56 from second-order statistics (of which 24 were generated from gray level co-occurrence matrices [GLCM], 16 were generated from gray level run length matrices [GLRLM], and 16 were generated from gray level size zone matrices [GLSZM]) using the Python PyRadiomics module (version 3.0.1; Computational Imaging and Bioinformatics Lab, Harvard Medical School, Boston, MA, USA).

The conventional machine learning algorithms are generally efficient in terms of computational power and memory through a feature selection process. Moreover, model performance is improved by removing the features that adversely affect learning in the process. Feature selection methods are generally divided into three types: filter methods, wrapper methods, and embedded methods. Each method selects a suitable subset of features for classification.

The filter method ranks each feature by calculating the relevance of features based on performance evaluation metrics and leaves a specified number of features in the highest order. The wrapper method is used to find the best feature subset while repeating the training via the algorithm. The embedded method internally selects a subset of features from the machine learning algorithm. The filter method selects features independent of the machine learning algorithm. Thus, the selected features do not affect the final classifier. The wrapper method has a high probability of overfitting and is time consuming. However, the embedded method is more accurate than the filter method because it selects features based on a machine learning algorithm (e.g., the wrapper method) and is likewise advantageous in terms of computational aspects [17]. Therefore, this study used an embedded method that takes advantage of each filter method, including the wrapper method.

The conventional machine learning algorithms used in the current study were logistic regression (LR), support vector machines (SVM), random forests (RF), and extreme gradient boosts (XGB). LR is an algorithm that uses regression to classify data by predicting the probability that data will be included in a particular category as a value between 0 and 1 [18]. SVM is an algorithm for defining the optimal decision boundary to categorize data using support vectors, meaning that the data point is near the decision boundary [19]. RF is a tree-based algorithm that draws conclusions by collecting classification results from multiple trees constructed through training [20]. XGB is likewise a tree-based algorithm. In the classification process comprising multiple decision trees, weights are provided for incorrect answers through the gradient of the loss function to predict results. Rapid training is possible through parallel processing [21]. The features selected from each algorithm are listed in Table 1.

### 2.4. Classification Based on Deep Learning

Among deep learning methodologies, CNN-based deep learning (extracting features using convolution operations) has recently been applied in medical imaging, showing high performance [22,23]. VGGnet is one of the deep learning networks designed to investigate the relationship between network depth and accuracy. VGGnet uses fewer parameters as compared with previously proposed convolution networks. Specifically, VGG uses a 3 × 3 convolution filter for all layers and obtains a superior performance by implementing a deeper network structure. VGGnet generally shows good performance and has been applied to many networks [24]. ResNet is a structure in which the output of the previous layer is connected to the input of the subsequent layer through a skip connection in order to enable learning using a deeper network structure [25]. In this study, we used four CNN-based deep learning models (VGG16, VGG19, Resnet50, and Resnet152) to investigate the most effective model for classifying and differentiating normal veins and thrombosis containing veins as well as analyzing the obtained results according to the depth of the network. The experiments were performed in Python 3.6.10 (Python Software Foundation, Wilmington, DE, USA) using Keras 2.2.5 frameworks (Keras Global Limited, London, UK) on a Ubuntu 14.04 operating system (London, UK) with two NVIDIA Tesla P100 graphics processing units (GPUs; NVIDIA Corporate, Santa Clara, CA, USA) and 512 GB of random access memory (RAM).

### 2.5. Performance Evaluation

To evaluate the effectiveness of each algorithm, a performance evaluation was conducted using 324 test data that were not used for training. We calculated true positives (TP), false positives (FP), false negatives (FNs), and true negatives (TNs). Sensitivity, specificity, and accuracy were calculated according to Equations (1)–(3) specified below. Receiver operating characteristic (ROC) curves were estimated using these sensitivity and specificity values. Algorithm performances were compared according to area under the curve (AUC) values derived from ROC curves.

(1)
$$\mathrm{Sensitivity}=\frac{\mathrm{TP}}{\mathrm{TP}+\mathrm{FN}}$$


(2)
$$\mathrm{Specificity}=\frac{\mathrm{TN}}{\mathrm{TN}+\mathrm{FP}}$$


(3)
$$\mathrm{Accuracy}=\frac{\mathrm{TP}+\mathrm{TN}}{\mathrm{TP}+\mathrm{TN}+\mathrm{FP}+\mathrm{FN}}$$


## 3. Results

We evaluated the performance of each model by applying fivefold cross-validation to assess model robustness. The results, including associated 95% confidence intervals, are shown in Table 2. The highest values are shown in bold. Based on AUC values, the algorithm with the highest performance was identified as the deep learning-based VGG16 network (0.982 ± 0.014), followed by the VGG19 model (0.981 ± 0.013). The VGG19 model added layers to the same model. Among the machine learning algorithms, XGB showed the highest performance (0.969 ± 0.005), with a difference of 0.007 as compared with VGG16. Figure 3a–c shows the ROC curves, AUC values, and standard deviations for each model.

Table 3 and Figure 3c shows a comparison between the average performances of machine learning-based algorithms and the average performances of deep learning-based algorithms. The deep learning algorithms showed a mean AUC of 0.939 (±0.043), a mean sensitivity of 0.897 (±0.057), a mean specificity of 0.894 (±0.028), and a mean accuracy of 0.890 (±0.045). The machine learning algorithms showed mean AUCs of 0.966 (±0.008), 0.917 (±0.025), 0.917 (±0.009), and 0.931 (±0.031), respectively. As demonstrated in Table 2, the algorithm with the highest overall performance was the deep learning-based VGG16 model. The average algorithm performance was higher within the machine learning models. Additionally, the performances of machine learning algorithms did not meaningfully differ according to the classifier when standard deviation was considered. However, in the case of the deep learning models, we observed a substantial difference in performance depending on the model.

## 4. Discussion

In this study, we evaluated 659 patients who underwent CT venography in order to detect DVT and investigate quantitative differences in performance with respect to automated classification of DVT using conventional machine learning and CNN models. As shown in Table 2 and Figure 3c, the algorithm that demonstrated the highest performance based on AUC values was the deep learning-based VGG16 model. CNN-based models, which differ from conventional machine learning methods necessitating manual steps with respect to feature extraction and selection, extract meaningful feature maps for training from convolutional layers. CNN demonstrates high performance through fitting a model using many parameters extracted through this process and has proven its validity within the current study. The VGG16 model based on CNN obtained an AUC value that was 0.007 higher than the XGB algorithm (which showed the highest performance among the conventional machine learning algorithms). Therefore, an appropriate amount of data (in terms of efficacy for deep learning) was used in this study. We determined that the deep learning-based algorithm extracted features more effectively with respect to classification as compared with manual feature extraction and feature selection based on statistical texture analysis. However, as shown in Figure 3b, the CNN-based ResNet model exhibited a statistically significantly lower performance. This result was likely due to the size of the image used as well as the deep structure of the Resnet-based model. Because the blood vessel area is quite small, the image size of the initial data was likewise small. Therefore, we judged that the high-dimensional features extracted from the deep layer were hindered in classifying the test data because of the information generated during the up-sampling process.

Conventional machine learning algorithms showed an overall high classification performance with an average AUC of 0.95 or more. We analyzed the features used for classification within the current study. In conventional machine learning algorithms, all algorithms select for the median, the median value of the gray level intensity, skewness, and the measured value of the asymmetry of the histogram via first-order statistics. Two features are predicted as having the most statistically significant impact on classifying the two categories. Features are ranked in the top 10 in order of importance for all four algorithms. The tree-based models (i.e., RF and XGB) with relatively high performances have the following features in common: mean (the mean values of gray level intensity), root mean squared (the square-root values of the mean of all the squared intensity), gray level variance (the variance of intensities for the gray level zones), and zone percentage (the ratio of the number of gray level zones and the number of connected voxels). Specifically, the GLCM-based cluster shade feature showed the highest feature importance for both models and had the most effective influence on classification in the current study. Additionally, in comparing performance differences according to classifier type, tree-based RF and XGB models and deep learning-based VGG models (i.e., nonlinear classifiers) showed higher performance compared to LR and SVM (i.e., linear classifiers). This indicates that our data are more suitable for nonlinear classifiers.

This study had several limitations. First, this retrospective study only considered data from a single medical center, leading to the possibility of selection bias. With the initiative towards prospective studies within academic medicine as well as pharmaceutical initiatives, we need to be prepared to accept AI in clinical practice. Second, the modest number of normal cases evaluated via CT venography hindered the training and validation of the CNN within the current study. Nevertheless, we considered patients with DVT in the clinical setting in order to minimize the spectrum bias. Third, we did not assess performance using coronal or sagittal reformatted CT images. Combining several CT images may improve performance. This prospective methodology should be investigated in the future.

In summary, we evaluated conventional machine learning and CNN-based methods for classifying DVT using a large dataset of CT venography in the current study. We found that CNN models classified DVT on serial CT images more effectively, especially via the VGG16 model and showed more accuracy in distinguishing DVT with a slightly higher AUC value as compared with the other AI algorithms. Our findings guide future research directions and will ultimately inform medical guidelines.

## Figures and Tables

**Figure 1 diagnostics-12-00274-f001:**
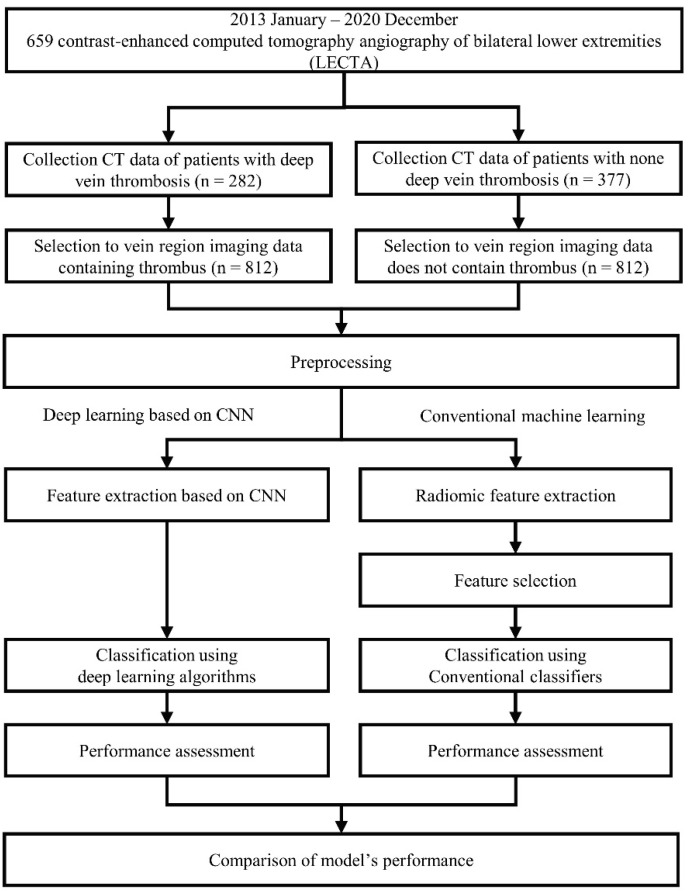
Flowchart of deep vein thrombosis (DVT) classification using deep learning models based on convolutional neural networks (CNN) and conventional machine learning.

**Figure 2 diagnostics-12-00274-f002:**
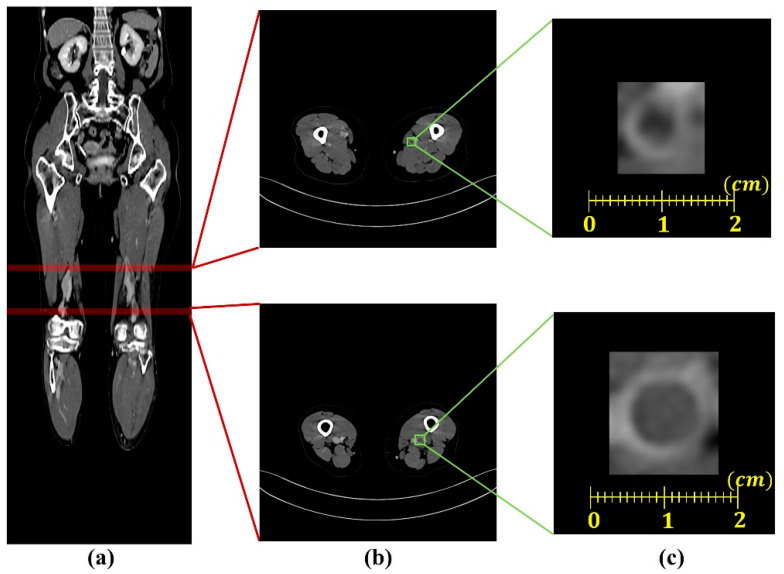
Example of generating the vein region image containing thrombus: (**a**) a contrast-enhanced lower extremity computed tomography venography image in coronal view; (**b**) computed tomography venography images containing thrombus; (**c**) generated vein region image from (**b**).

**Figure 3 diagnostics-12-00274-f003:**
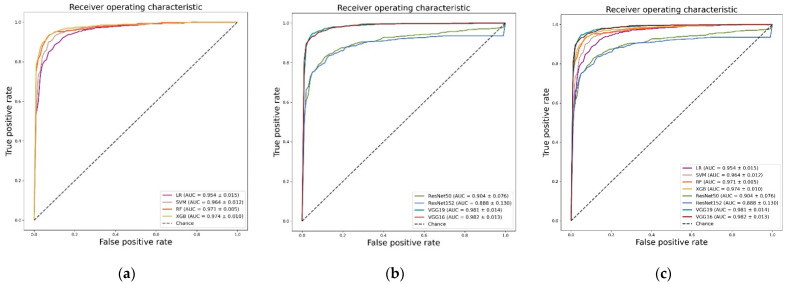
Receiver operation characteristic (ROC) curves: (**a**) ROC curves of machine learning algorithms; (**b**) ROC curves of deep learning algorithms; (**c**) ROC curves of all algorithms.

**Table 1 diagnostics-12-00274-t001:** Selected features from machine learning algorithms.

	Feature Number	Features
Logistic regression	20	First order: 10th percentile, 90th percentile, entropy, maximum, median, minimum, skewness GLCM: cluster tendency, correlation, inverse difference normalized, inverse variance, sum entropy GLRLM: gray level non uniformity, long-run low gray level emphasis, run entropy, run variance GLSZM: gray level non uniformity normalized, large area high gray level emphasis, size zone non uniformity normalized, small area emphasis, zone entropy
Support vector machine	20	First order: 10th percentile, 90th percentile, entropy, maximum, median, robust mean absolute deviation, skewness GLCM: cluster shade, correlation, inverse difference normalized, informational measure of correlation 1, informational measure of correlation 2, joint entropy, maximum probability, sum entropy GLRLM: high gray level run emphasis GLSZM: high gray level zone emphasis, large area high gray level emphasis, size zone non uniformity normalized, zone entropy
Random forest	19	First order: mean, median, root mean squared, skewness GLCM: autocorrelation, cluster prominence, cluster shade, difference variance, joint average, joint energy, maximum probability, sum average GLRLM: gray level variance, high gray level run emphasis, long-run high gray level emphasis, short–run high gray level emphasis GLSZM: large area high gray level emphasis, small area high gray level emphasis, zone percentage
Extreme gradient boost	18	First order: mean, median, range, root mean squared, skewness, uniformity GLCM: cluster shade, contrast, inverse difference normalized, informational measure of correlation 2, maximum probability GLRLM: gray level non uniformity normalize, gray level variance, high gray level run emphasis, run length non uniformity normalized, short run emphasis GLSZM: gray level variance, zone percentage

GLCM, gray level co-occurrence matrix; GLRLM, gray level run length matrix; GLSZM, gray level size zone matrix.

**Table 2 diagnostics-12-00274-t002:** Comparison of the mean values for each model’s AUC, sensitivity, specificity, and accuracy.

	Mean AUC (±95% CI)	Mean Sensitivity (±95% CI)	Mean Specificity (±95% CI)	Mean Accuracy (±95% CI)
LR	0.954 (±0.002)	0.879 (±0.002)	0.902 (±0.004)	0.889 (±0.002)
SVM	0.964 (±0.001)	0.913 (±0.001)	0.919 (±0.003)	0.915 (±0.001)
RF	0.969 (±0.001)	0.932 (±0.002)	0.921 (±0.002)	0.926 (±0.001)
XGB	**0.975 (±0.002)**	**0.945 (±0.002)**	**0.926 (±0.002)**	**0.935 (±0.002)**
VGG16	**0.982 (±0.001)**	**0.956 (±0.003)**	0.916 (±0.005)	0.934 (±0.003)
VGG19	0.981 (±0.002)	0.950 (±0.004)	**0.926 (±0.005)**	**0.935 (±0.003)**
Resnet50	0.904 (±0.008)	0.858 (±0.007)	0.859 (±0.010)	0.849 (±0.006)
Resnet152	0.888 (±0.014)	0.825 (±0.006)	0.873 (±0.012)	0.841 (±0.008)

AUC, area under a receiver operating characteristic curve; CI, confidence interval; LR, logistic regression; SVM, support vector machine; RF, random forest; XGB, extreme gradient boost. The highest performance values are shown in bold.

**Table 3 diagnostics-12-00274-t003:** Comparison of the mean performance values of deep learning and machine learning algorithms.

	Deep Learning Algorithms	Machine Learning Algorithms
Mean AUC (±SD)	0.939 (±0.043)	**0.966 (±0.008)**
Mean sensitivity (±SD)	0.897 (±0.057)	**0.917 (±0.025)**
Mean specificity (±SD)	0.894 (±0.028)	**0.917 (±0.009)**
Mean accuracy (±SD)	0.890 (±0.045)	**0.931 (±0.031)**

AUC, area under a receiver operating characteristic curve; SD, standard deviation. The highest performance values are shown in bold.

## Data Availability

The datasets generated and/or analyzed during the current study are available from the corresponding author upon reasonable request.

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
