# Peer review of "Comparison between Deep Learning and Conventional Machine Learning in Classifying Iliofemoral Deep Venous Thrombosis upon CT Venography"

_diagnostics, 2022, doi:10.3390/diagnostics12020274_

Round 1
Reviewer 1 Report
This paper aims to compare the performance of conventional ML to CNN in detecting DVT on CT venography. Population for validation is not clear and the overall number of scans used is unclear in order to draw conclusions.
Introduction
– sentence 1 – please insert reference.
-second to last sentence first paragraph – please insert reference even if it is known.
Clearly state the objective/aim of the study
Methods and Materials
Revise the flow chart – “Start” is not acceptable. Maybe can integrate with number of images – identifying vessels (which) and then comparing processes
Number of images: CT images from 282 patients with DVT and 377 “controls” = 659. How does the number of images become 812? Please explain or correct. Are these data points or features chosen? Scans with or w/o contrast?
Explain “the remaining data points… “(what was the total).
Why data points and not number of images? Each image has a number of data points--
How many CT scans were used for validation? How many scans (from which you derived data points) were used for testing? The number in validation must be greater than that of testing.
2.3: “Statistics based on first-order and second-order statistics are used for classifying images” – based on what?
“From a total of 1,624 images…” – please explain number of images which is not consistent with what declared previously (see also confusion regarding data points)
-please re-write feature selection paragraph. Not clear.
This section it refers to machine learning algorithms As does 2.4. Make sure to explain the difference.
2.4 – explain what VGG stands for
Results:Model robustness is high, but must display the validation cohort data and the testing cohort data with “n” values and describe correctly in results.Figure 2 – b and c are the same graphMake sure there are no discussion sentences in the results
Author Response
Response to Reviewer 1 Comments
We appreciate your insightful feedback on ways to strengthen our paper. We tried to actively reflect your opinions and revise the manuscript based on your advice.
Introduction
Point 1. sentence 1 – please insert reference.
We agree with your advice. We have inserted reference as you suggested (p. 1; line 37).
Point 2. second to last sentence first paragraph – please insert reference even if it is known.
We agree with your advice. We have inserted reference as you suggested (p. 1; line 43).
Point 3. Clearly state the objective/aim of the study
Thank you for your comment. We have additionally provided aim of the study (p. 2; lines 24 - 26).
Methods and Materials
Point 4. Revise the flow chart – “Start” is not acceptable. Maybe can integrate with number of images – identifying vessels (which) and then comparing processes.
We appreciate your suggestion. We revised the flow chart reflected this comment (p. 3; Figure 1).
Point 5. Number of images: CT images from 282 patients with DVT and 377 “controls” = 659. How does the number of images become 812? Please explain or correct. Are these data points or features chosen? Scans with or w/o contrast?
Thank you for your question. In generating the image data for classification, when the thrombus did not extend along the z-axis, it was judged as another thrombus. Furthermore, in the axial view, there are some case with more than one vein containing a thrombus. The below figures show example of some case with one patient has multiple vein region images with thrombus.
Therefore, we generated more than one vein region image data from each patient. We added a new Figure 2 (p. 4) to further illustrate the generating image data in this study.
That is, in this study, one vein region image was generated for one thrombus. Therefore, each image was generated for multiple thrombus from the same patient.
We hope these below figures can help to explain the generating the dataset.
Figure 1. Example of image which has vein region containing thrombus more than one.
Figure 2. Example of CT data which has vein region containing thrombus more than one.
Point 6. Explain “the remaining data points… “(what was the total).
Thank you for your suggestion. We rewrote the sentence to be more explicit (p. 3; lines 17-18).
Point 7. Why data points and not number of images? Each image has a number of data points—
There is mistake selection of word. We agreed with your comment and changed the word “data points” to “images” throughout the script (p. 3; lines 12-18).
Point 8. How many CT scans were used for validation? How many scans (from which you derived data points) were used for testing? The number in validation must be greater than that of testing.
Thank you for your necessary question. The number of train, validation and test set are written in section 2.2 (p. 3; lines 16-17).
However, we used the same number of validation and test set in this study because we applied 5-fold cross validation and split the train, validation, test as 3: 1: 1 ratio. According to the general artificial intelligence experiment in medical imaging based on the best our knowledge, the train, validation, and test set data ratio usually set to 3:1:1 (5-fold) or 8:1:1 (10-fold) [1-2]. As our knowledge, the validation set is used for enhancing the robustness of model and avoiding overfitting by validating the model performance during training. If there's something we're missing out on, we'd appreciate it if you could give us advice about this.
[1] Schlemper, J.; Oktay, O.; Schaap, M.; Heinrich, M.; Kainz, B.; Glocker, B.; Rueckert, D., Attention gated networks: Learning to leverage salient regions in medical images. Med Image Anal 2019, 53, 197-207.
[2] Yu, T.; Canales-Rodriguez, E. J.; Pizzolato, M.; Piredda, G. F.; Hilbert, T.; Fischi-Gomez, E.; Weigel, M.; Barakovic, M.; Bach Cuadra, M.; Granziera, C.; Kober, T.; Thiran, J. P., Model-informed machine learning for multi-component T2 relaxometry. Med Image Anal 2021, 69, 101940.
Point 9. 2.3: “Statistics based on first-order and second-order statistics are used for classifying images” – based on what?
We had a mistake of translation the sentence. We have rewritten the sentences (p. 4, lines 11-13) to be more in line with our intention. We hope that the edited sentences clarify.
Point 10. “From a total of 1,624 images…” – please explain number of images which is not consistent with what declared previously (see also confusion regarding data points)
Thank you for your comment. Please see point 5 above.
Point 11. -please re-write feature selection paragraph. Not clear.
We appreciate your advice. We revised the feature selection paragraph (p. 4, lines 20-23) to establish a clear meaning.
Point 12. This section it refers to machine learning algorithms As does 2.4. Make sure to explain the difference.
We agree with you and have incorporated this suggestion throughout our paper by removing the sentence “Deep learning is a machine-learning algorithm” and adding the definition of conventional machine learning and deep learning in this study for explaining the difference (p. 2; lines 33-37). The added sentences are more in line with your suggestion. We think these changes now better than before. We hope that you agree.
Point 13. 2.4 – explain what VGG stands for
Thank you for your comment. However, “VGG” is the name of deep learning network [1]. We changed the word “VGG” to “VGGnet” throughout the manuscript to avoid confusion (p. 6; lines 5-9).
[1] Simonyan, K.; Zisserman, A., Very deep convolutional networks for large-scale image recognition. arXiv preprint arXiv:1409.1556 2014.
Results
Point 14. Model robustness is high, but must display the validation cohort data and the testing cohort data with “n” values and describe correctly in results. Figure 2 – b and c are the same graphMake sure there are no discussion sentences in the results
Please see point 5 above for the explanation about the number of data (p. 3; lines 16-18).
We mistake to attach the figures. Thank you for notice. We changed the Figure 3-c (before Figure 2-c, p.7) to the correct version. Moreover, we mentioned with discussion sentences about each Figure 3 in the results section and discussion section (p.7, 8).
Reviewer 2 Report
Dear Authors, below are my comments about the submitted manuscript.
- I would spend more words in the Introduction about CNN.
- In the initial part of the Materials and Methods section you stated: “All participants provided their written informed consent prior to participation.”, that is in contrast with the back matter of the article, Informed Consent Statement, where you indicated: “Not applicable”. Please clarify,
- For comprehensiveness, I would add some representative figure about the CT images evaluated.
Author Response
Thank you for your interest in our study and helpful comments. We reflected your suggestions that help our paper to clarify our goal.
Point 1. I would spend more words in the Introduction about CNN.

We appreciate the reviewer’s suggestion. We added sentence about CNN in the introduction (p. 2; lines 11-15).
Point 2. In the initial part of the Materials and Methods section you stated: “All participants provided their written informed consent prior to participation.”, that is in contrast with the back matter of the article, Informed Consent Statement, where you indicated: “Not applicable”. Please clarify,
We thank the reviewer for pointing this out. It’s our mistake. We have changed the sentences (p. 2; lines 31-32).
Point 3. For comprehensiveness, I would add some representative figure about the CT images evaluated.
Thank you for providing these insights. We reflected this comment by adding the Figure 2 (p. 4) to further illustrate the data used in this study.